# Applications of Stimuli-Responsive Hydrogels in Bone and Cartilage Regeneration

**DOI:** 10.3390/pharmaceutics15030982

**Published:** 2023-03-18

**Authors:** Xiaoqi Ni, Xin Xing, Yunfan Deng, Zhi Li

**Affiliations:** The State Key Laboratory Breeding Base of Basic Science of Stomatology (Hubei-MOST) and Key Laboratory of Oral Biomedicine Ministry of Education, School and Hospital of Stomatology, Wuhan University, Wuhan 430079, China

**Keywords:** stimuli-responsive, hydrogels, bone, cartilage, regeneration

## Abstract

Bone and cartilage regeneration is an area of tremendous interest and need in health care. Tissue engineering is a potential strategy for repairing and regenerating bone and cartilage defects. Hydrogels are among the most attractive biomaterials in bone and cartilage tissue engineering, mainly due to their moderate biocompatibility, hydrophilicity, and 3D network structure. Stimuli-responsive hydrogels have been a hot topic in recent decades. They can respond to external or internal stimulation and are used in the controlled delivery of drugs and tissue engineering. This review summarizes current progress in the use of stimuli-responsive hydrogels in bone and cartilage regeneration. The challenges, disadvantages, and future applications of stimuli-responsive hydrogels are briefly described.

## 1. Introduction

Bone and cartilage defects resulting from trauma, infections, tumors, or congenital conditions are common and pose substantial clinical burdens. Although bone tissue has a certain “self-healing” ability, it does not demonstrate this innate ability beyond “critical” defects [1,2]. Cartilage is an avascular and poorly cellularized tissue with a limited capacity for self-renewal. Even minor chondral defects may necessitate surgical intervention [3]. Thus, the repair of bone and cartilage defects is a considerable challenge for clinicians. 

Autogenous/allogeneic bone grafts or inert metal/ceramic implants are the main treatment options for bone defects. Autogenous bone grafts are considered the gold standard for treating minor bone defects. Moreover, various surgical techniques, such as microfracture and allogenic and autologous cartilage transplantation, have been used to repair articular-cartilage defects [4,5]. However, such treatments have the potential to cause more severe issues, including graft rejection, disease spread, infection, pain, bone nonunion, and osteonecrosis [6,7,8,9,10], and the long-term survival of grafts remains uncertain [11,12]. 

Tissue engineering is a novel strategy for promoting tissue repair and regeneration by combining related supporting cells, three-dimensional scaffolds, and bioactive factors [13]. The complete regeneration of damaged tissue is achieved by the in vitro or in vivo synthesis of a biological matrix with the same properties as the original healthy tissue [1,14]. This synthesis provides a new strategy for treating bone and cartilage defects in clinical environments. The most significant aspect of tissue engineering is the construction of scaffolds that provide structural support for cell migration, adhesion, proliferation, and differentiation, followed by the creation of a suitable extracellular matrix-like growth environment for cell growth [15,16]. The ideal scaffold for bone and cartilage tissue engineering should have good mechanical strength, biocompatibility, controlled degradation, good porosity, and an interconnected porous environment. These properties can provide a microenvironment that supports cell migration, adhesion, differentiation, and proliferation [17].

Hydrogels are materials with a 3D hydrophilic network consisting of cross-linked polymer chains that can absorb large amounts of water or solvents without degradation [18,19], allowing cells to adhere and differentiate onto the hydrogels. They can simulate the natural tissue environment and support the healing of defects [16,20]. The high water content of hydrogels makes them porous, highly permeable, and flexible, similar to natural tissue, facilitating the rapid diffusion of oxygen and nutrients within the scaffold [21,22,23]. The mechanical properties of hydrogels, such as hardness, pore size, and degradation rate, can be altered by changing the polymer composition, molecular weight, crosslinking density, and polymerization conditions [7,24]. Moreover, the structure and composition of hydrogels are similar to those of an extracellular matrix (ECM), which can provide a similar external environment for the growth of host cells and tissues [22,25], and drug release can be controlled by changing the pore size over a long time [24,26]. Based on many studies on the application of hydrogels in bone and cartilage tissue regeneration, it has been proved that their numerous properties meet the requirements of ideal scaffolds for tissue engineering, so they are widely considered to be one of the most suitable biomaterials [27].

To date, various hydrogels have been developed and used. In recent years, growing interest in personalized pharmacotherapy and precision medicine has prompted the development of smart responsive-hydrogels. Smart responsive-hydrogels can respond to exogenous stimuli (light, electricity, pressure, and magnetic fields) or endogenous stimuli (temperature, enzymes, pH, ion concentration, and ROS (reactive oxygen species)) and undergo abrupt changes in their physical properties and macroscopic alterations [25,28,29,30,31]. Physiological changes can provoke the controllable release of biologically active molecules, cells, or drugs. In general, the actions of different stimulating factors will cause hydrogels to undergo reversible or irreversible changes in their chemical structures and physical properties [22], such as degradation, shrinking or swelling, sol–gel transition, competitive binding, and self-assembly [29]. Smart responsive-hydrogels with unique features are being developed for use in controlled drug delivery [24,32,33], tissue engineering [34,35], biosensors [14,36], wound dressing [37,38,39], and cancer treatment [40,41].

Numerous studies have revealed that damage to bone or cartilage tissue, including damage caused by fractures or inflammation, causes changes in the microenvironment of the damaged site and abnormal cell activities, such as changes in enzymes [42,43,44,45,46,47,48,49,50,51,52,53], pH [54,55,56], hypoxia [57,58,59], temperature, and redox reactions [60,61,62,63]. These characteristic changes can act directly as internal triggers to provoke smart responsive-hydrogels for drug release. In addition to these internal stimuli, external triggers can be applied to synthesize smart/responsive hydrogels for bone and cartilage tissue engineering (Figure 1). The application of stimuli-responsive hydrogels facilitates on-demand release, ensuring that growth factors or cells arrive at specific defective tissues and providing mechanical and biological support for bone and cartilage defects [29,64]. 

This literature review summarizes the application of stimuli-responsive hydrogels in bone and cartilage regeneration, mainly focusing on polymer-based crosslinking stimulus-responsive hydrogels. The strategies covered in this review are summarized in Table 1.

## 2. Different Types of Stimuli-Responsive Hydrogel

### 2.1. Enzyme-Responsive Hydrogels

Enzymes and enzymatic reactions have recently emerged as promising trigger motifs for developing new responsive polymers. Enzymes are vital biocatalysts since all biological metabolic processes of organisms require the participation of enzymes. In addition, the enzyme catalytic reaction has high substrate specificity and selectivity, and catalytic reaction conditions are mild (water medium, neutral or weak acid, and weak base environment). Enzyme-responsive hydrogels can bind the enzyme directly to the polymer via covalent bonding or encapsulation, and they can also interact directly with enzyme-reactive hydrogel polymers. For developing functional enzyme-reactive biomaterials, the following components must be involved: (1) recognition elements or substrate mimics that enzymes can recognize; (2) the enzyme-substrate reaction must result in changes in the structure of materials; (3) the enzymes should have easy access to anchored substrates that can significantly affect the kinetics of enzyme-catalyzed reactions; (4) therapeutic molecules, such as drugs, growth factors, and cells, can be linked to polymeric materials through physical and chemical methods [101,102,103]. Under pathological conditions, such as rheumatoid arthritis, osteoarthritis, and bone injury, high levels of MMPs (MMP-1, MMP-2, MMP-3, MMP-7, MMP-8, and MMP-9) can be detected at the site of injury. Hydrogels have the potential for use in minimally invasive cell delivery and thus have great prospects in tissue regeneration therapy.

Numerous molecules involved in the physiological process of bone regeneration have gradually become known. Bone morphogenetic proteins (BMPs) are well-studied and commonly used. BMPs can effectively induce the osteogenic differentiation of mesenchymal stem cells (MSCs) [104]. Coletta et al. [65] developed two different bioactive materials, elastin-like recombinamers (ELRs) containing BMP-2 or a cell adhesion peptide motif (Arg-Gly-Asp: RGD), to form a BMP-2 loaded extracellular-matrix-like hydrogel. The material degrades via an enzymatic reaction through the addition of an elastase-sensitive domain into the ELR, so the scaffold should not be removed after bone healing. The scaffold can control the release of BMP-2, which is beneficial in repairing bone defects. Furthermore, RGD demonstrates remarkable cell adhesion. Aisenbrey et al. [69] developed a biodegradable human mesenchymal stem cell (hMSC)-loaded hydrogel using 8-arm PEG norbornene, thiolated chondroitin sulfates, mono-thiol-containing cell adhesion peptide (CRGDS), and an MMP7-sensitive peptide in the presence of photoinitiator Igracure 2959 under 352 nm light (Figure 2A). Adding a cartilage-derived extracellular matrix into the synthetic process of a hydrogel creates a physiological microenvironment for cartilage formation. After several weeks of hMSC culture coupling with the hydrogel degradation, the expression of cartilage-specific genes such as SOX9, COL2A1, and ACAN increases. The formation and deposition of aggrecan and collagen II can be observed at six weeks. The MMP7-responsive hydrogel can promote the formation of macro-cartilage tissue that consists entirely of sGAGs, aggrecan, and collagen II. Holloway et al. [66] prepared a maleimide-modified hyaluronic acid (MAHA)-based enzyme-sensitive hydrogel containing a cell adhesion peptide (RGD) and a matrix metalloproteinase sensitive peptide (Figure 2B). This hydrogel loads and transports stromal-derived factor-1α (SDF-1α) and BMP-2. BMP-2 release can be observed after implanting the hydrogel into a rat skull defect. The rate is proportional to the rate of hydrogel degradation, and new bone tissues are detected. Hsu et al. [67] proposed a new biomaterial design strategy for synthesizing a cathepsin-K-sensitive poly (ethylene glycol) diacrylate hydrogel modified by acryloyl-PEG-succinimidyl carbonate containing cathepsin-K-sensitive peptide. Cathepsin-K is significantly expressed in osteoclasts during bone resorption. Therefore, it will induce the specific degradation of a hydrogel, which is expected to be utilized in bone remodeling. Aziz et al. [68] created an MMP-sensitive poly (ethylene glycol) hydrogel containing an MMP-sensitive peptide and adhesion peptide (RGD) to promote the deposition of bone extracellular matrix and support osteoblast-to-osteocyte differentiation of IDG-SW3 cells compared to undegradable hydrogels (Figure 2C).

### 2.2. pH-Responsive Hydrogels

A pH-sensitive hydrogel is a synthetic polymer that responds to different pH levels. When the pH of the microenvironment changes, the hydrogel modifies its mechanical properties through the protonation/deprotonation mechanism. As a result, alternating the chemical structure of polymer hydrogels can be controlled via the pH level [26,105]. Rogina et al. [72] proposed a pH-responsive chitosan-hydroxyapatite-based hydrogel using sodium bicarbonate (NaHCO_3_) as a gelling agent to allow rapid gelling of CS and hydroxyapatite via physical crosslinking. After co-culture with mouse fibroblasts, the cells showed good viability and proliferation. This hydrogel proved to be promising for application as a cell carrier. Zhao et al. [70] developed a pH-triggered, self-assembled nanoparticle hydrogel scaffold by incorporating carboxymethyl chitosan (CMCh) and amorphous calcium phosphate (ACP). When the pH was adjusted from 11 to 7.5, using glucono–δ–lactone (GDL) as the acidifier into 1 mg/mL dispersion of hybrid nanoparticles, the surface of nanoparticles comprised positively or negatively charged particles, and the nanoparticles shrunk and formed aggregates. Self-assembly of the nanoparticles was induced to form a stable gel and realize the transition from liquid-like to solid-like. The in vivo behavior of CMCh-ACP hydrogel as a BMP-9 carrier has been further studied. The BMP9-CMCh-ACP hybrid-gel-treated group had a denser and more mature bone-like matrix and higher osteogenic and chondrogenic marker expression than other groups. These results suggest that CMCh-ACP hybrid gel provides an ideal scaffold for effective BMP9-induced bone formation. The pH-sensitive self-assembly peptide (SAP) P_11_-4 (CH_3_COQQRFEWEFEQZQQNH_2_) can be converted to low-pH and physiological salt concentration fibers. Still, it will be a monomer at high pH and low salt concentration [106], and this process can be reversibly controlled by adjusting the pH level. Saha et al. [71] combined P_11_-4 (CH_3_COQQRFEWEQQFEQQNH_2_) hydrogel with human dental pulp stromal cells to promote bone tissue repair in vivo, and the P_11_-4 treated group exhibited significant bone regeneration and almost complete defect healing. The results show that the self-assembled peptide would be a suitable scaffold for acellular bone tissue engineering.

### 2.3. Temperature-Responsive Hydrogels

Temperature-sensitive hydrogels can be easily mixed with low-viscosity polymer solutions at room temperature and transported to the defect site by injection. The hydrogels are usually in a sol state at room temperature (25 °C). When heated to body temperature (37 °C), they will transform into a gel, thus realizing in situ gelation of the internal environment [22,107]. A range of proteins and peptides can be used to design temperature-responsive hydrogels.

MSCs can differentiate into many other cells to form new tissues, and bone marrow-derived stem cells (BMSCs) are widely used in bone and cartilage tissue engineering. Liao et al. [73] proposed an injectable, thermo-responsive hyaluronic acid-g-chitosan-g-poly (N-isopropyl acrylamide-g-poly) (HA-CPN) hydrogel constructed as a scaffold to induce differentiation and osteogenesis of rASCs. Chitosan-based hydrogels as cell carriers and materials for defect repair represent an area of in-depth research in bone and cartilage tissue repair. Chitosan is obtained via deacetylation of chitin and is composed of b-(1/4)-2-amido-2-deoxy-D-glucan (glucosamine) and b-(1/4)-2-acetamido-2-deoxy-D-glucan (acetyl glucosamine) units [108]. Chitosan can be dissolved in acidic solutions with a pH less than or equal to 6.5, leading to the protonation of its amine groups [65,67]. The sol–gel transition occurs when β-glycerophosphate is added to neutralize the positive charge during the preparation process so that chitosan can exist as a soluble form in the neutral solution. In addition, the prepared material rises from room- to physiological-body-temperature (37 °C) [109,110]. This feature allows the incorporation of growth factors and cells into the chitosan sol to form a hydrogel at the defect site as a repairing material. Hoemann et al. [77] and Naderi-Meshkin et al. [81] used β-glycerophosphate (β-GP), chitosan (CS), and hydroxyethyl cellulose (HEC) to prepare a biocompatible and biodegradable hydrogel scaffold for cartilage regeneration, considering the thermo-sensitive property of CS/β-GP hydrogel. Hoemann et al. [77] successfully used CS-GP-HEC hydrogel as a carrier and protector for the transportation of primary chondrocytes and a scaffold for the formation of newly born tissue without rapid degradation. Furthermore, Naderi-Meshki et al. [81] utilized transforming growth factor β3 (TGF-β3) to induce hMSC differentiation. Their study aimed to determine the chondrogenic differentiation capacity of encapsulated hMSCs. hMSCs can remain biologically active for 28 days when injected into the defect site. Simultaneously, TGF-β3 promotes the conversion of human mesenchymal stem cells to cartilage phenotypes, indicating that chitosan-beta glycerophosphate-hydroxyethyl cellulose (CH-GP-HEC) is likely to be used to repair cartilage tissue defects. Polyethylene glycol is one of biomedicine’s most widely used polymers due to its biological inertness, versatility, and hydrophilicity. Its hydrophilic property allows it to bind with hydrophobic, biodegradable polymers such as polylactic acid (PLA) or polycaprolactone acid to form amphiphilic polymers [111]. Yeon et al. [112] synthesized a temperature-sensitive poly (ethylene glycol)-b-poly (L-alanine) (PEG-L-PA) hydrogel to explore its potential as a three-dimensional medium for an adipose-tissue-derived stem cell (ADSC) culture. ADSCs were cultured in PEG-L-PA hydrogel. At 37 °C, in vivo and in vitro experiments have confirmed that ADSCs have a strong potential for differentiation into chondrocytes. Zhang et al. [78] encapsulated BMSCs in a thermo-responsive tri-segment copolymer poly (lactide-co-glycolide)-block-poly (ethylene glycol)-block-poly (lactide-co-glycolide) (PLGA-PEG-PLGA) hydrogel for cartilage defect repair. At physiological temperature, the hydrogel demonstrated reversible sol–gel transition. After 12 weeks of implanting the BMSC-loaded PLGA-PEG-PLGA hydrogel in a rabbit articular cartilage defect, the hydrogel+BMSC group showed new cartilage tissue with similar biology and mechanical properties to normal cartilage. Liu et al. [79] introduced L-phenylalanine into a polyalanine-based thermosensitive hydrogel to prepare poly (L-alanine-co-L-phenylalanine)-block-poly (ethylene glycol)-block-poly (L-alanine-co-L-phenylalanine) (PAF-PEG-PAF) triblock copolymers. By adjusting the concentration of phenylalanine groups, the physical and mechanical properties of gels, including the temperature of the sol–gel transition, the critical gelling concentration, and the pore size of the gels, were changed, which also had an impact on the adhesion and proliferation of BMSCs. After 12 weeks, the rabbit patellar cartilage defect treated with PAF-PEG-PAF (EG_91_A_28_F_9_) hydrogel scaffold implantation had more smooth, continuous, thick, and transparent new cartilage tissue than the other groups. Makvandi et al. [80] also constructed a temperature-sensitive hydrogel composed of β-tricalcium phosphate, hyaluronic acid, and corn silk extract-nanosilver (CSE-Ag-NPs). The sol-gelation temperature of the gel was close to the physiological temperature. Ag nanoparticles have antibacterial properties, especially against Gram-positive and Gram-negative bacteria, and a strong promotion effect on the osteogenic differentiation of coated MSCs. The Zn-CS/β-GP hydrogel prepared by Niranjan et al. [74] was transformed into a solid gel at a physiological temperature (37 °C), promoting osteoblast differentiation in the osteoblast culture medium. Concurrently, the presence of Zn conferred the material bacteriostatic action against Gram-positive and Gram-negative bacteria. Segredo-Morales et al. [75] introduced a new type of injectable thermo-sensitive poloxamine (T-1307) hydrogel reinforced by alginate and cross-linked with calcium chloride (CaCl_2_), containing 17-β-estradiol, BMP-2, and plasma rich in growth factors (PRGF). 17-β-estradiol was encapsulated in poly (D, L-lactide) (PLA-S) microspheres and locally applied to induce bone regeneration in OP (osteoporosis) female rats using PLGA (poly (lactic-co-glycolic acid))-PVA (polyvinyl alcohol) microspheres to load BMP-2.

Alginate addition resulted in hard gel formation at low temperatures, prolonging the existence of the hydrogel. The sustained release of BMP-2 and 17-β-estradiol effectively improved bone defect repair in OP (osteoporosis) rats, but the experiments demonstrated that PRGF did not improve bone repair [75]. Similarly, Li et al. [76] proposed a well-controlled powder sintering technique to develop a porous Ti6AI4V scaffold with high porosity and large pore size. The thermosensitive chitosan thioglycolic acid (CS-TA) hydrogel was used as the carrier of rhBMP-6 microspheres. Microspheres loaded with rhBMP-2 were transferred to scaffold pores at 37 °C to prepare a Ti6AI4V/rhBMP-2 hydrogel composite material, realizing the stable and sustained release of rhBMP-2 and effectively promoting bone regeneration.

The application of dual-stimulation-responsive hydrogels in bone and cartilage tissue engineering has gained increasing attention. The two most common and controllable stimulating factors in the biological environment are pH and temperature. pH/temperature-sensitive synthetic polymers are mainly synthesized by combining pH-sensitive polypeptides and temperature-sensitive polymers [113]. Kim et al. [82] attached a pH-sensitive sulfamethazine oligomer (SMO) to both ends of a thermo-sensitive poly (e-caprolactone-co-lactide)-poly(e-caprolactone-co-lactide)-poly(PCLA-PEG-PCLA). Under physiological conditions (pH = 7.4, 37 °C), the synthesized block copolymer formed a stable gel quickly and had a remarkable encapsulation efficiency for stem cells and growth factors. Hydrogels containing hMSCs and BMP-2 were implanted in the subcutaneous tissues of nude mice for up to seven weeks. The newly mineralized tissues were revealed to have high alkaline phosphatase activity. Chitosan is a natural polymer, and chitosan hydrogels are not affected by pH and temperature variations. Ding et al. [83] created the double responsive groups pH-responsive C6-OH allyl-modified CS (OAL-CS) and temperature-responsive poly (N-isopropyl acrylamide) (PNIPAM), which were incorporated into chitosan. Under UV irradiation, OAL-CS and PNIPAM formed a hydrogel network structure via “thiol-ene” click chemistry (Figure 3). 

The swelling rate of a hydrogel can be regulated by changing pH, temperature, and the proportion of OAL-CS to PNIPAM in the system (Figure 4). An in vivo experiment demonstrated that the hydrogel has no toxicity to BMSCs. OAL-CS/PNIPAM hydrogel can be considered an intelligent biomaterial for bone tissue engineering.

### 2.4. ROS-Responsive Hydrogels

Recently, biomaterials have been designed to contain redox-sensitive components (such as disulfide, tellurium, and diselenide bonds) that can be broken down in the presence of reducing agents (such as glutathione and dithiothreitol) to control material degradation and the release of drugs, growth active factors, and cells. The most common redox pairing in a responsive crosslinking system is glutathione (GSH) and its corresponding oxidized species (glutathione disulfide). Glutathione can regulate reducing conditions by forming and destroying disulfide bonds, which are mainly dependent on glutathione/NADPH+. Furthermore, the cytosol contains substantially higher glutathione (2–10 mM) than the extracellular environment [114,115,116]. Glutathione is considered a key element in determining the redox environment of cells since ROS-sensitive hydrogels can exhibit their benefits at high intracellular glutathione levels. In addition, inflammatory states, often accompanied by the production of reactive oxygen species, are one of the body’s defense mechanisms against invasion by pathogens and other foreign bodies. An increase in glutathione synthesis can be detected in abnormal reactive oxygen microenvironments. Therefore, utilizing glutathione state variation as a stimulus to construct redox-sensitive hydrogels is considered feasible. 

Based on the abnormal responses of the ROS system in the inflammatory response, a core–shell redox-responsive nano-hydrogel was constructed to control the release of BMP-2 [84]. The inner core region of the nano-hydrogel consists of polyethylene oxide (PEO) and BMP-2. The shell comprises poly (ethyl lactone) (PCL) and a redox-responsive c-6A PEG-PCL nanogel with an -S-S- bond (Figure 5). 

BMP-2 is regulated by the degradation of hydrogels in response to changes in glutathione concentrations both in vitro and in vivo. In vitro experiments revealed that the controlled delivery of BMP-2 via redox-sensitive nanofibers exhibited good osteogenesis potential in mandible defect repair (Figure 6). In response to a glutathione (GSH)/oxidized glutathione (GSSG) level change, the authors of [84] also synthesized another redox-sensitive hydrogel composed of six-arm poly (ethylene glycol)–poly(ε-caprolactone)-3,3′-dithiodipropionic acid gels (6A PECL-SS) and six-arm poly (ethylene glycol)–poly (ε-caprolactone)-acryloyl (6A PECL-AC). The breakage and formation of disulfide bonds can trigger a reversible modification of the cross-linked structure, resulting in the reversible transformation of the elastic module. Using that hydrogel as the substrate for rBMSC culture displayed great osteoinduction under various redox stimulations [85]. In addition, Yang et al. [86] synthesized a disulfide-containing, 4-armed-PEG-based scaffold loaded with rhBMP-2 that can be controlled by disulfide bond breakage in the presence of reductive GSH or other oxidative species.

### 2.5. Magnetic-Responsive Hydrogels

Magnetically responsive hydrogels are usually prepared using three main preparation methods: in situ precipitation, blending, and grafting. Magnetic components, such as iron oxide-based magnetic nanoparticles (MNPs) (Fe_3_O_4_) and cobalt ferrite nanoparticles (CoFe_2_O_4_), are integrated into the polymer matrix to prepare hydrogels that can respond to a magnetic field [117,118,119]. Furthermore, biologically active factors can be combined with MNPs to be directed and intelligently transported to specific areas under the action of an external magnetic field. The magnetic field can be used as an external stimulus to induce specific biomechanical signals, thereby regulating human cell behaviors, such as proliferation, differentiation, or apoptosis [120]. Hydroxyapatite (HA) has good biocompatibility due to its similar inorganic components to natural bone tissue and is a popular bone replacement biomaterial. However, HA has poor cell adhesion, slow crawling, and other shortcomings [121]. Huang et al. [87] developed a new type of magnetic nanocomposite hydrogel through the ultrasonic dispersion method and freeze–thaw crosslinking molding using polyvinyl alcohol (PVA) nano-hydroxyapatite (n-HA) and magnetic nanoparticles (Fe_2_O_3_) as raw materials; BMSCs grew evenly on the surface of the magnetic nanocomposite hydrogel, and the growth rate of BMSCs was significantly high. Hou et al. [88] found that the osteoblast adhesion and proliferation rate substantially improves when the concentration of m-nHA (Fe_2_O_3_ coating with nano-hydroxyapatite) increases. In general, a magnetic field (alternating or static magnetic field) can be used to control biological behaviors. To further study whether magnetic stimulation can be applied to bone regeneration, Yuan et al. [89] constructed a magnetic stimulation in vitro platform for bone formation mechanisms, where osteoblasts (MG-63) and magnetic iron oxide nanoparticles (IONPs) were loaded into natural collagen hydrogels to develop a 3D model. Under a static magnetic field (SMF), the combined application of SMFS+IONPs significantly improved the production and mineralization of alkaline phosphatase, the proliferation of MG-63 cells, and the expression of BMP-2 and BMP-4, RUNX2, and osteocalcin (OCN), compared with the control group. However, its applicability in vivo requires further research. 

It is critical to ensure adequate blood supply at the bone defect site. Stromal vascular fraction (SVF) is a heterogeneous cell system including fibroblasts, endothelial cells, macrophages, hemocytes, pericytes, and different mesenchymal stem cell types with robust angiogenic ability [122]. Filippi et al. [90] incorporated magnetic nanoparticles (MNPs) and SVF cells extracted from human adipose tissue with polyethylene glycol (PEG)-based hydrogel to develop a new type of three-dimensional magnetic nano-hydrogel with both osteogenic and vasculogenic properties (Figure 7A,B). The content of CD31^+^ + endothelial cells in the magnetically actuated gels was higher, and more elongated capillary-like structures formed (Figure 7C). Furthermore, Abdeen et al. [91] presented an adjustable magnetic hydrogel matrix by embedding functional carbonyl iron (CI) particles in a polyacrylamide hydrogel. CI particles can enhance gel elasticity in a high magnetic field without affecting cell growth. When MSCs were implanted in the magnetic hydrogel matrix, their angiogenesis and osteogenic differentiation potential increased. The authors of [91] concluded that magnetic stimuli played an essential role in initiating osteogenesis signals.

Bone tissue is sensitive to mechanical stimulation. Appropriate mechanical stimulation is essential for maintaining bone structure and function. Bone cells are considered the predominant mechanosensing cell type in bone tissue, and their apoptosis, proliferation, and differentiation can be regulated by activating molecular signaling pathways [123,124]. 

The cell adhesion peptide Arg-Gly-Asp (RGD) is regarded as an important factor in promoting cell adhesion and enhancing osteogenic differentiation [125], whereas the TREK-1 channel is mechanically sensitive [126]. Studies have revealed that TREK-1 can be activated by mechanical stimulation to guide the osteogenic differentiation of mesenchymal stem cells and promote the expression of osteogenic and chondrogenic genes [127]. Magnetic nanoparticles can be used as a channel through which to transmit mechanical stimulation. In ref. [92], functionalized magnetic nanoparticles (MNPs) were used to label human bone marrow mesenchymal stem cells (hBMSCs) as mechanically gated TREK1K+ channels or cell surface mechanoreceptor Arg-Gly-Asp (i.e., integrin) RGD binding domains, before transferring them to a collagen-based hydrogel coated with BMP-2. In in vivo experiments, hMSCs loaded with RGD-MNPs or TREK1-MNPs showed more mineralization in chicken fetal femurs, with TREK1 being more effective in promoting bone formation in both models. In combination with BMP-2, the hydrogel with MNP-labeled hMSCs showed more significant bone formation and mineralization. Therefore, magnetic nanoparticles can deliver mechanical stimulation to amplify the intracellular cascade effect of exogenous growth factors to optimize their therapeutic efficacy [92]. Magnetic hydrogels can also be used in cartilage tissue repair. Zhang et al. [93] introduced magnetic nanoparticles into a hydrogel using collagen type II, hyaluronic acid (HA), and polyethylene glycol (PEG) to prepare a magnetic nanocomposite hydrogel for cartilage tissue engineering. The hydrogel scaffold could be placed precisely at the cartilage defect. The adhesion density of BMSCs in the hydrogel was found to be significantly higher than that of control and gel groups without MNPs. The team also found that BMSCs could engulf MNPs and maintain their original activity and morphology. However, further research is required to determine whether MNPs can be broken down by lysosomes in cells and excreted from cells via exocytosis. BMPs promote cartilage formation and stimulate endochondral ossification [128]. Fan et al. [94] created nucleobase-paired chitosan/heparin composite nanogels to encapsulate Fe_3_O_4_ nanoparticles using Watson–Crick base pairing between thymine and adenine. The controlled release of BMP-2 was achieved under an external magnetic field. In addition, the magnetic nano-hydrogel loaded with BMP-2 promoted the proliferation and differentiation of MG-63 cells, indicating that this hydrogel has potential applications in cartilage and bone regeneration. However, the properties of magnetic particles, such as content, size, shape, and crystallinity, can affect the performance of hydrogels. A thorough evaluation should focus on the effects on organisms and hydrogel-loaded cells or biomolecules produced by external magnetic fields [119,129].

### 2.6. Photo-Responsive Hydrogels

Light is considered an external stimulus with various advantages, such as being low-cost and spatiotemporally controllable; it can also be used to modulate the properties of hydrogels [130]. A photo-responsive hydrogel was obtained by incorporating alginate-acrylamide hybrid gels (AlgAam) with ferric ions by Giammanco et al. [13]. Under visible light irradiation, the chemical composition and microstructure of Fe(III)-alginate (e.g., swelling properties and porosity) could be changed. A live/dead assay demonstrated that this hydrogel has good biocompatibility. The production of sulfated glycosaminoglycan (sGAG) produced by cartilage precursor cells (ATDC5) cultured on the photochemically treated hydrogels was significantly increased, and the amount of sGAG is an essential marker for evaluating cartilage formation. Platelet-rich plasma (PRP) contains several growth factors that facilitate wound repair and can be used to repair articular cartilage injuries [131]. Liu et al. [95] prepared an in situ photo crosslinked PRP hydrogel (HNPRP) by incorporating autologous PRP with a photo-responsive hyaluronic acid (HA-NB) (Figure 8A). In that study, based on the property that HA-NB can generate aldehyde groups under light irradiation (395 nm LED light), which could subsequently react with amino groups to achieve imine crosslinking, the HNPRP hydrogel precursor solution was able to rapidly undergo crosslinking reactions to accomplish in situ gelation. In vitro experiments demonstrated that HNPRP hydrogels were cytocompatible and had strong tissue adhesive ability that could promote the proliferation and migration of chondrocytes and BMSCs, which are essential for cell recruitment at the defect (Figure 8C,D). In addition, the results showed that HNPRP hydrogels could realize the controlled release of growth factors (Figure 8B). 

Metal-organic frameworks (MOFs), a new type of porous crystalline material, are produced through the formation of coordination bonds between metal ions and organic ligands. The synthesis conditions influence their overall structure and properties with the advantages of tunability, high specific surface area, and high porosity. They can adsorb or bind drugs on their surface via van der Waals forces, hydrogen bonds, coordination bonds, and electrostatic forces and can be used as a delivery system for tissue engineering applications [132,133]. Multiple MOFs have been combined with biological scaffolds that promote bone regeneration via delivering osteogenic drugs [54], promoting osteogenic differentiation of cells [134], and inducing matrix mineralization [135], proving that MOFs have great potential in bone tissue engineering. In one study [96], dexamethasone-loaded zeolitic imidazolate frameworks-8 (DZIFs) were incorporated into a hydrogel matrix of methacrylic polyphosphoester (PPEMA) and methacrylic gelatin (GelMA) through photo crosslinking. Dexamethasone was effectively delivered into the periodontal pocket, exerting antibacterial and anti-inflammatory effects and slowing down bone loss. Sun et al. [97] proposed an in situ photo crosslinked nanocomposite hydrogel for achieving a controlled release of cobalt (Co) ions. 2-ethylimidazole (eIm) was introduced into zeolitic imidazolate framework-67 (ZIF-67), which was then combined with gelatin methacrylate (GelMA) under UV light to form a GelMA@eIm/ZIF-67 hydrogel (Figure 9). The controlled release of Co ions could be achieved by changing the amount of linkers for a maximum of 21 days. Co ions promote angiogenesis in the early stage of bone formation. This composite hydrogel can be used as a drug release and proangiogenic/bone regeneration system for bone defect repair.

However, the effects of light intensity, duration, and reactive intermediates on the physiological behaviors of the organisms still need to be evaluated through experiments. The chemical and physical properties of hydrogels determine the photo-responsive properties of the hydrogels. The structure of the materials requires further optimization if fast, safe, and reversible reactions are to be obtained under light irradiation [136].

### 2.7. Electro-Responsive Hydrogels

Electro-responsive hydrogels are commonly composed of polysolvates, or they can be made electrically conductive by adding conductive particles, including metal nanoparticles and carbon-based materials [98,137,138]. Using an external electric field is an effective strategy to activate/regulate reactions, which undergo a series of changes such as corrosion, swelling, deswelling, or bending in the presence of electric fields [137,139]. The intensity, direction, and duration of the electrical stimulus, the composition of the polymer, the solution pH, and the ionic strength all affect the hydrogel’s electrosensitive behaviors [140,141,142]. Electro-responsive hydrogels should have good response times, deformation, and memory function, and low cost and low power consumption. Otherwise, their application will be limited to a great extent. So far, electro-responsive hydrogels have been widely used in several intelligent device fields, for instance, sensors, membrane separation devices, and drug delivery systems [140]. Previous studies have demonstrated that electro-responsive hydrogels can mediate cell adhesion, proliferation, migration, and growth from different tissues, including cardiac [143], skeletal [144], muscular [145], and neural [146] tissues. Rahimi et al. [99] incorporated fibrin into an anionic polymer acidacrylic acid to form a novel electrosensitive hydrogel, and after providing 2 h of pulsed electrical stimulation, porcine smooth muscle cells (pSMCs) exhibited significant cell migration on hydrogels and uniform distribution of cells throughout the structure caused by alternating swelling and deswelling of the hydrogel (Figure 10A,B). Hydrogels with porous structures and high-water content can load bioactive molecules or drugs into polymers that release bioactive molecules or drugs in vivo for therapeutic purposes in response to external electrical stimulation [147].

The controlled-release properties of electro-responsive hydrogels can be modified through the following mechanisms: electric-field-induced swelling or dehydrated shrinkage in the hydrogel; electrophoresis of charged drugs, electrophoretic effects, or erosion of the hydrogel caused by an electric field [148]. Qu et al. [100] synthesized a biodegradable electro-responsive hydrogel composed of dextran and an electroactive aniline trimer using hexamethylene diisocyanate as a crosslinking agent for drug delivery. The addition of an aniline trimer provided the hydrogel with good electrical conductivity. The hydrogel was able to release more drugs rapidly when external electrical stimulation was applied, compared to no extra voltage being used. The design of the hydrogel realizes a stepwise or continuously modulated and precise drug release via the “on-off” mechanism of the external electrical field (Figure 10C,D).

A variety of biomaterials based on graphene and its derivatives, including graphene nanogrids [149], graphene-oxide-based hydrogels [150], and graphene oxide foams [146], have great application prospects in ontogenesis, tissue regeneration, cancer therapy, and drug delivery [151,152,153], exhibiting excellent biological properties, electrochemical properties, high surface area, excellent mechanical strength, strong hydrophilicity, and high adsorption capacity [151,154,155], along with effects on stem cell differentiation and proliferation [156,157]. Among them, graphene oxide (GO) and its reduced form, reduced graphene oxide (rGO), are the most widely studied graphene-derived forms to date [151]. The stimulus-responsive property of polymers is known to be achieved by adding suitable fillers to the matrix, and rGO with high electrical conductivity can significantly enhance the electrical conductivity of polymers [158]. It has been shown that rGO can control drug release in response to the presence or absence of external stimuli, demonstrating the potential of rGO as a stimulus-responsive matrix [159]. MacKenna et al. [98] added reduced graphene oxide (rGO) to a conductive hydrogel composed of jeffamine polyetheramine and polyethylene glycol diglycidyl ether (PEGDGE) and assayed the release kinetics of the hydrogel, adding methyl orange (MO) as a model drug. With the addition of rGO, the passive release of MO from the hydrogel was significantly reduced, and the mechanical strength of the hydrogel increased. The release rate and dose of MO can be adjusted by changing the rGO content, electric field polarity, and amplitude. 

However, there is still much controversy over the application of electric fields to the human body, and the hydrogel may be toxic to cells due to external electric fields.

## 3. Conclusions and Perspectives

Stimuli-responsive hydrogels have a wide range of potential future applications in bone and cartilage tissue engineering. These stimuli-responsive materials perform their specific function according to the status of the body’s disease physiological changes, the natural physiological characteristics of the organism, or external stimulation. Hydrogels will degrade, expand, or release therapeutic drugs and growth factors in response to specific stimuli, to maintain local concentrations of molecules, cells, and drugs at disease sites. Hydrogels have a natural advantage in bone and cartilage regeneration due to their superior chemical and physical properties. 

Efforts have been made to prove the superiority of stimuli-responsive hydrogels. Before moving toward clinical applications and commercialization, some practical problems still need to be addressed. Inflammation is a critical factor that promotes increased expression of MMPs, pH alteration, and disruption of the intracellular redox system. Therefore, intelligent, responsive systems developed for inflammatory diseases in bone or cartilage tissue may not accurately ensure the controlled release of all therapeutic substances when other systemic inflammation is present, resulting in misplaced release or off-target damage [29]. The precise identification of the target site may not be achieved by relying on a single stimulation pattern. A dual or multiple-responsive hydrogel can be considered an effective strategy to lead a “real-time” response to the targeting sites and acquire the precise release of the encapsulated therapeutics, aiming to provide a higher effective dose. Variations in microenvironment diseases are conducive to designing responsive hydrogels that only act in specific pathological states, avoiding premature degradation, off-target release, or initial burst release. Hence, future perspectives on the biomarkers of various diseases will further facilitate the improvement of responsive hydrogel. The variability of individual disease characteristics hinders the clinical translation of bio-responsive systems. For responsive hydrogels modulated under external stimuli, the stimuli factors can be artificially controlled, while the effect of hydrogels in response to internal stimuli can be impacted by differences in individual physiological states. We should promote the personalized design of intelligent, responsive hydrogels and develop a specific “response” mode for each individual to make the treatment process more precise and efficient. This requires an accurate understanding of each patient’s disease characteristics and physiological changes. The key point is that biological signals can be artificially altered; for instance, through signal amplification and precise detection.

The current research demonstrates that physiological-stimulus-mediated changes are passive and irreversible, and their responsive process cannot be modulated accompanying the disease progression [160]. Unlike conventional smart-responsive hydrogels, programmable hydrogels can reversibly change their properties and functions, ensuring the on-demand release of the therapeutic payload and, in particular, achieving modifiability of the dose and rate of encapsulated substances [103]. Biocompatibility is a major issue for all types of biomaterials. Therefore, the biological/toxicological effects of the materials and reaction intermediates should be thoroughly evaluated before applying smart, responsive hydrogels in bone and cartilage tissue engineering. The materials should possess good biocompatibility and non-cytotoxicity to ensure that they do not affect the normal physiological activities of the organism. For responsive hydrogels that rely on external stimuli, the adverse effects of external stimuli, including electric and magnetic fields, on the organism should be evaluated, and further efforts are needed to explore their safe usage dose. Some crosslinking agents used in the preparation of hydrogels are toxic to cells, GF, and tissues, thus limiting their expanded application in tissue engineering, and multiple synthesis methods or the development of less toxic crosslinking agents should be considered. In addition, the biodegradability of the responsive hydrogel should be optimized. The degradation rate should be consistent with the rate of tissue regeneration as much as possible, thus providing reliable and robust support for tissue regeneration. Most hydrogels are soft materials with a much lower elastic modulus than natural bone tissue, so the application of hydrogels in bone tissue engineering has been limited compared to other tissues. The mechanical strength of hydrogels can be enhanced by adjusting the number of crosslinks, changing the type of monomer, or adding other materials [161].

In summary, smart, responsive hydrogels are promising for applications in modulating cell behavior, drug delivery, and tissue engineering. However, many challenges remain.

## Figures and Tables

**Figure 1 pharmaceutics-15-00982-f001:**
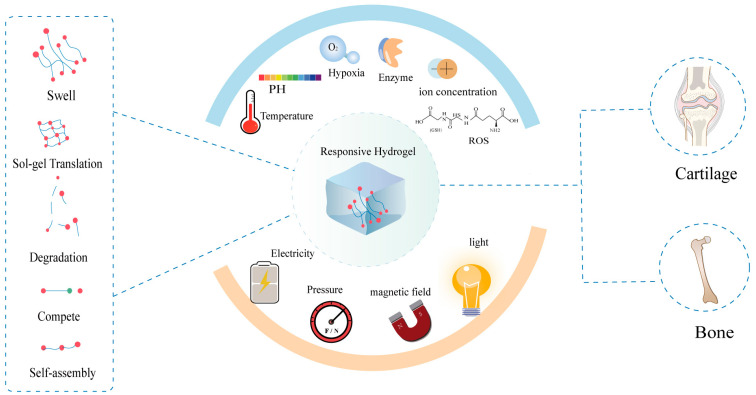
Stimuli-responsive hydrogels and their biological application.

**Figure 2 pharmaceutics-15-00982-f002:**
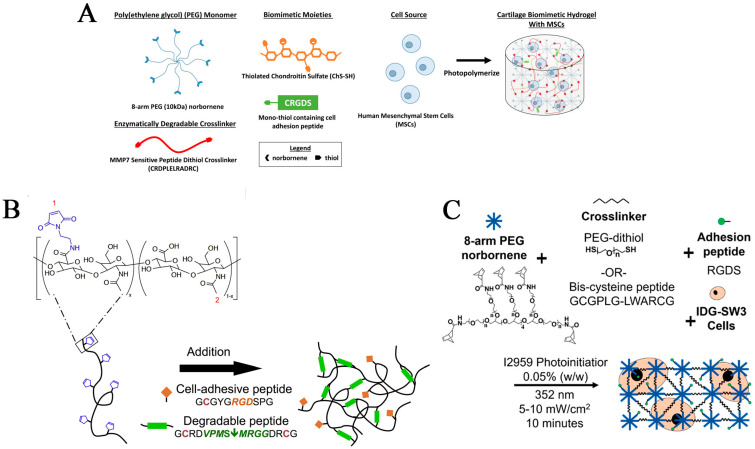
(**A**) Schematic illustration of the preparation of a human mesenchymal stem cell (hMSC)-encapsulated, MM7-responsive cartilage mimetic hydrogel. Reprinted with permission from [69]. (**B**) Maleimide-functionalized hyaluronic acid (MaHA) and the formation of cell-adhesive MaHA hydrogels. Cell-adhesive MaHA hydrogels were formed via an addition reaction between maleimides of MaHA and thiols of cysteine groups within cell-adhesive peptides (RGD) or MMP-sensitive degradable peptides (VPMS↓MRGG). Reprinted with permission from [66]. (**C**) Schematic of hydrogel formation and cell encapsulation through a photoclickable reaction. Reprinted with permission from [68].

**Figure 3 pharmaceutics-15-00982-f003:**
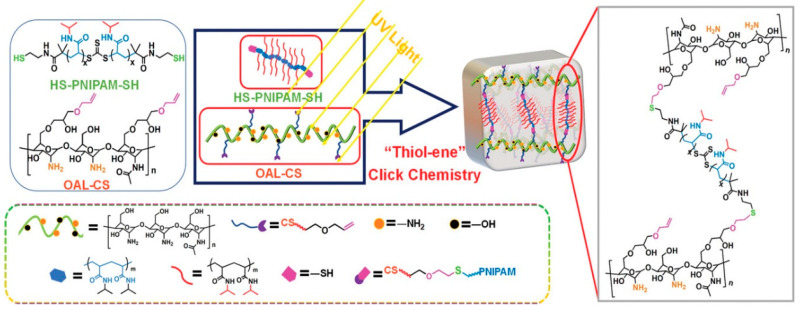
The design and preparation of dual pH- and thermo-responsive OAL-CS/PNIPAM hydrogel via thiol-ene click chemistry. Reprinted with permission from [83].

**Figure 4 pharmaceutics-15-00982-f004:**
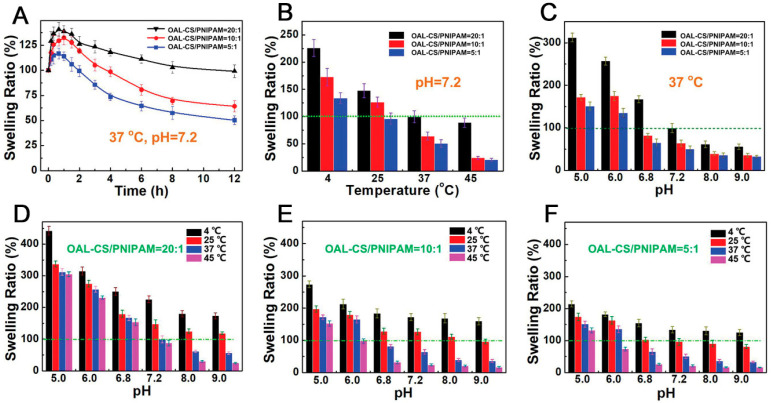
(**A**) Swelling ratios of different OAL-CS/PNIPAM hydrogels at pH 7.2 and 37 °C. (**B**) Swelling ratios of different OAL-CS/PNIPAM hydrogels in pH 7.2 PBS at different temperatures. (**C**) Swelling ratios of different hydrogels at different pH values at 37 °C. (**D**) Changes in swelling ratio in OAL-CS/PNIPAM = 20:1 hydrogel via the effect of temperature and pH. (**E**) Changes in swelling ratio in OAL-CS/PNIPAM = 10:1 hydrogel via the impact of temperature and pH. (**F**) Changes in swelling ratio in OAL-CS/PNIPAM = 5:1 hydrogel via the effect of temperature and pH. Reprinted with permission from [83].

**Figure 5 pharmaceutics-15-00982-f005:**
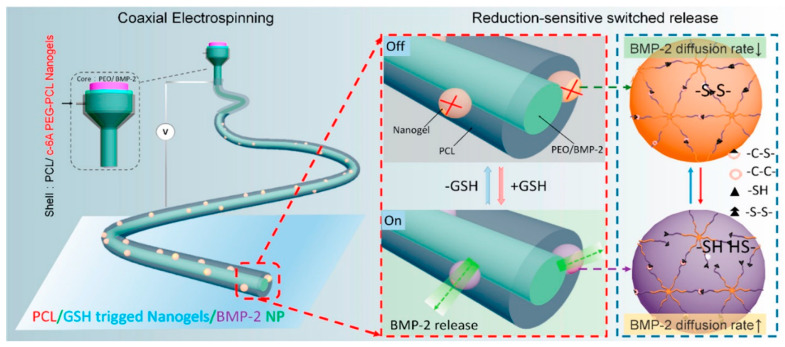
Schematic illustration of the fabrication of a nanogel-in-nanofiber device with coaxial electrospinning and the redox-responsive release with the nanogel. Reprinted with permission from [84].

**Figure 6 pharmaceutics-15-00982-f006:**
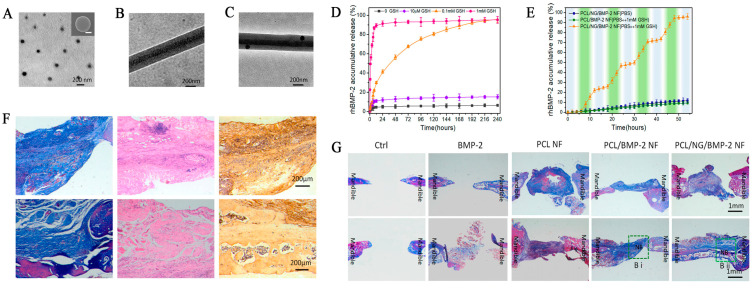
(**A**) TEM and SEM (the inset) images of nanogels (**B**,**C**); TEM images of nanogel in nanofiber; (**D**) redox-responsive release behavior of BMP-2 in different GSH contents at 37 °C; (**E**) stepwise response of nanofibers on increasing GSH concentration; (**F**) high magnification of Masson, HE, and immunohistochemical staining of the bone defect; (**G**) histomorphometric results of the bone defect area (NB: new bone). Reprinted with permission from [84].

**Figure 7 pharmaceutics-15-00982-f007:**
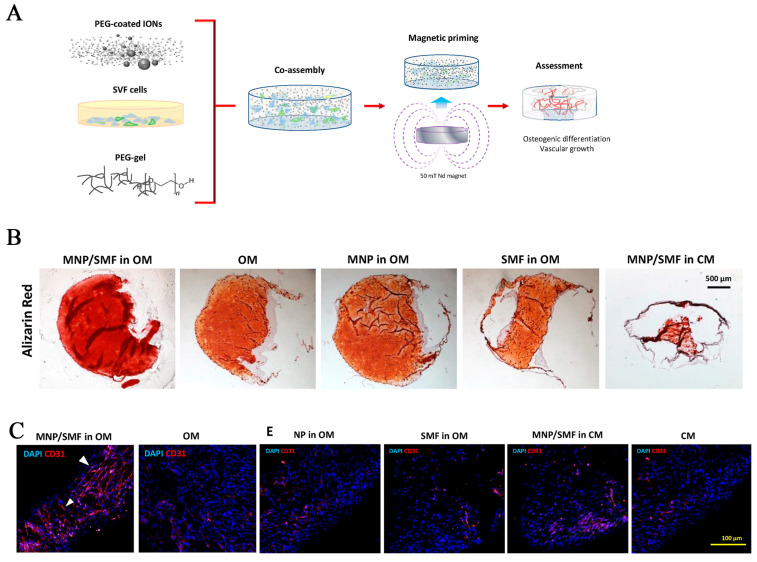
(**A**) The formation of magnetic hydrogels via co-assembly of cells, particles and PEG gels, and their stimulation under an external magnetic field. (**B**) Alizarin red staining was performed to evaluate the level of calcium deposition (in red). (**C**) Percentage of CD31+ cells cultured for 3 weeks by confocal microscopy to assess angiogenic capacity. (Osteogenic medium, OM, or culture medium, CM). Reprinted with permission from [90].

**Figure 8 pharmaceutics-15-00982-f008:**
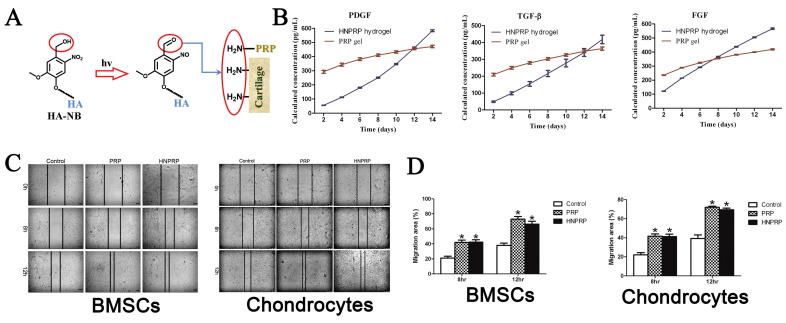
(**A**) Synthesis process of HNPRP hydrogel. (**B**) Release kinetic curves of growth factors (n = 3) released from HNPRP hydrogel or thrombin-activated PRP gel. (**C**) Images of BMSCs and chondrocytes migrating to the scratch area under the presence of HNPRP hydrogel or thrombin-activated PRP gel. (**D**) Statistical data of migration area for BMSCs and chondrocytes (* *p* < 0.05 compared with control group). Reprinted with permission from [95].

**Figure 9 pharmaceutics-15-00982-f009:**
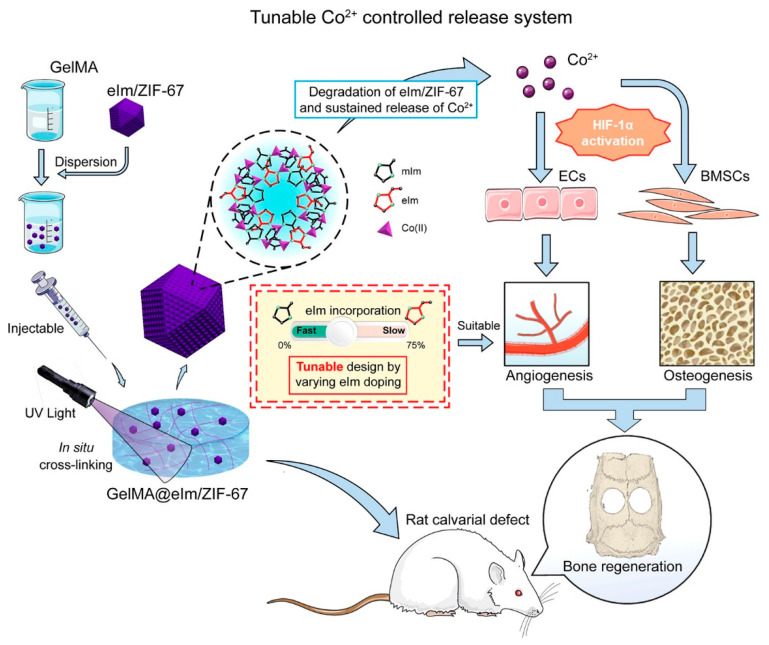
Schematic illustration of the GelMA@elm/ZIF-67 nanocomposite hydrogel, the delivery of Co ions, and the mechanism for angiogenesis and osteogenesis. Reprinted with permission from [97].

**Figure 10 pharmaceutics-15-00982-f010:**
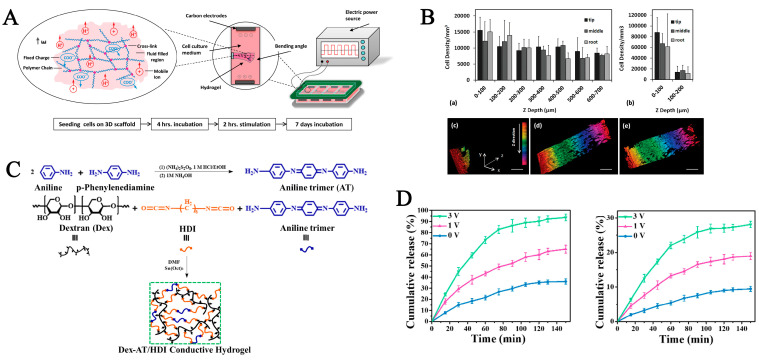
(**A**) Schematic of polyelectrolyte hydrogel bending when electrical stimulation is performed; the hydrogel shrinks at the anode and expands at the cathode. (**B**) Cell distribution in the hydrogels under stimulated condition or unstimulated condition. The distribution of pSMCs in PAA/fibrin hydrogels under stimulated condition by electrical field (**a**) or non-stimulated condition (**b**) in the long axis of the gel with a distinction in three regions. (**c**) No electric field was applied. (**d**) Electrical stimulation without verapamil addition or (**e**) with verapamil treatment. Reprinted with permission from [99]. (**C**) Schematic of aniline trimer (AT) and Dex-AT/HDI hydrogel. (**D**) Drug release of dexamethasone and indomethacin in phosphate buffer pH = 7.4 at different electric potentials. Reprinted with permission from [100].

**Table 1 pharmaceutics-15-00982-t001:** Stimuli-responsive hydrogels in bone and cartilage tissue engineering.

Stimuli	Components	Growth Factor/ Cell/Drug	Synthesis Methods	Gelation Time	Application	Ref.
Enzyme	Elastin-like recombinamers elastase-sensitive domain	BMP-2, RGD	Crosslinking	/	Bone regeneration	[65]
	MMP7-sensitive peptide, maleimide-modified hyaluronic acid, RGD	SDF-1α and BMP-2	Crosslinking	5 min	Bone regeneration	[66]
	Poly(ethyleneglycol) diacrylate (PEGDA), cathepsin-K-sensitive peptide GGGMGPSGWGGK (GPSG)	/	Crosslinking	1 h	Selective degradation	[67]
	Polyethylene glycol	ID-SW3	Crosslinking	10 min	Cell differentiation	[68]
	PEG norbornene, thiolated chondroitin sulfates, GRGDS, MMP7-sensitive peptide	hMSCs	Crosslinking	8 min	Cartilage regeneration	[69]
pH	Carboxymethyl chitosan, amorphous calcium phosphate	BMP-9	Self-assembly	30 min	Bone regeneration	[70]
	(SAP)P_11_-4(CH_3_COQQRF EWEFEQQQNH_2_)	HDPSCs	Self-assembly	/	Bone regeneration	[71]
	Chitosan, hydroxyapatite	Fibroblasts	Crosslinking	4 min	Cell growth	[72]
Temperature	Hyaluronic acid-g-chitosan-g-poly (N-isopropylacrylamide-g-poly)	rASCs	Crosslinking	/	Osteoblastic differentiation, ECM mineralization	[73]
	Zn, chitosan, β-glycerophosphate	MSCs	Crosslinking	5 min	Osteoblast differentiation of MSCs	[74]
	PoloxamineT-1307, alginate, calcium chloride	17β-estradiol, BMP-2, PRGF	Crosslinking	15 min	Bone regeneration	[75]
	Ti6AI4V, chitosan thioglycolic acid	BMP-2	Crosslinking	2.62 ± 0.87 min	Bone regeneration	[76]
	Poly(ethylene glycol) -b-poly(L-alanine)	/	Crosslinking	10 min	Chondrogenic differentiation of ADSCs	[77]
	Poly(lactide-co-glycolide)-block-poly(ethylene glycol) -block-poly(lactide-co-glycolide)	BMSCs	Polymerization	/	Chondrogenic differentiation of BMSCs and cartilage repair	[78]
	L-Phenylalanine,poly(L-alanine-co-L-phenylalanine)-block-poly(ethylene glycol) -block-poly(L-alanine-co-L-phenylalanine)	BMSCs	Crosslinking	/	Cartilage repair	[79]
	β-tricalcium phosphate, hyaluronic acid corn silk extract-nanosilver	MSCs			Osteogenic differentiation of MSCs	[80]
	β-glycerophosphate, chitosan, hydroxyethyl cellulose	Primary articular chondrocytes	Crosslinking	Few minutes	Cartilage regeneration	[77]
	β-glycerophosphate, chitosan, Hydroxyethyl cellulose	TGF-β3, hMSCs	Crosslinking	20 min	Chondrogenic differentiation of hMSCs	[81]
	Sulfamethazine oligomer, Poly(e-caprolactone-co-lactide)-Poly(e-caprolactone-co-lactide)-poly	hMSCs, BMP-2	Crosslinking	/	Bone regeneration	[82]
	C6-OH allyl-modified chitosan, Poly(N-isopropyl acrylamide)	/	Crosslinking	60 s	Drug delivery	[83]
ROS	Poly-LRB-ethylene oxiPeo, Poly (ethyl lactone), redox-responsive c-6A PEG-PCL	BMP-2	Crosslinking, electrospinning	/	Controlled release, bone regeneration	[84]
	A mixture of six-arm poly (ethylene glycol)-poly (ε-caprolactone)-3,3′-dithiodipropionic acid gels, six-arm poly(ethylene glycol)-poly(ε-caprolactone)-acryloyl	/	Crosslinking	/	Bone regeneration	[85]
	Polyethylene glycol	rhBMP-2	Crosslinking	/	Controlled release, bone regeneration	[86]
Magnetic field	Polyvinyl alcohol, nano-hydroxyapatite, magnetic nanoparticles (Fe_2_O_3_)	BMSCs	Crosslinking, ultrasonic dispersion.	/	Cell growth, chondrogenic differentiation	[87]
	Nano-hydroxyapatite, poly(vinyl alcohol)	Osteoblasts	Freeze–thawing	/	Cell adhesion and proliferation	[88]
	Collagen, iron oxide nanoparticles	MG-63	Crosslinking, co-assembly	/	Cell proliferation, bone formation	[89]
	Polyethylene glycol	SVF cell	Crosslinking	/	Osteogenesis, vascularization	[90]
	Polyacrylamide, carbonyl iron	MSCs	Crosslinking, co-assembly	/	Osteogenesis, vascularization	[91]
	Collagen, RGD or TREK1K+	BMP-2, nanoparticle-labeled hMSCs	/	/	Bone formation	[92]
	Collagen type II, hyaluronic acid, polyethylene glycol	BMSCs	Crosslinking	/	Cell adhesion, magnetic guidance	[93]
	Chitosan, Heparin	BMP-2	Watson–Crick pairing, co-assembly	/	Cell viability, delivery of growth factors	[94]
Light	Alginate-acrylamide hybrid gels (AlgAam), ferric iron	ATDCs, BMSCs	Crosslinking	/	Cartilage formation	[13]
	Hyaluronic acid	PRP	Crosslinking	/	Proliferation and migration of BMSCs and chondrocytes	[95]
	Zeolitic imidazolate frameworks-8, methacrylic, polyphosphoester (PPEMA), methacrylic gelatin (GelMA)	Dexamethasone	Crosslinking	20 s	Drug delivery	[96]
	2-ethylimidazole (eIm), zeolitic imidazolate framework-67 (ZIF-67), gelatin methacrylate (GelMA)	Co-icons	Crosslinking	/	Drug delivery, vascularization, bone formation	[97]
Electrictiy	Jeffamine polyetheramine, polyethylene glycol diglycidyl ether (PEGDGE), rGO	Methyl orange	Crosslinking, co-assembly	/	Drug delivery	[98]
	Fibrin, acrylic acid	pSMC	Free-radical polymerization and crosslinking	/	Cell migration	[99]
	Dextran, aniline trimer, hexamethylene diisocyanate	/	Crosslinking		Drug delivery	[100]

## Data Availability

Data sharing not applicable—no new data generated.

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
