# Peer review of "Applications of Stimuli-Responsive Hydrogels in Bone and Cartilage Regeneration"

_pharmaceutics, 2023, doi:10.3390/pharmaceutics15030982_

Round 1

Reviewer 1 Report

Review of manuscript ID: pharmaceutics-2235762 entitled ‘Applications of Stimuli-Responsive Hydrogels in Bone and 2 Cartilage Regeneration’ by Xiaoqi Ni and colleagues.

The above manuscript is a review on bone and cartilage regeneration. This is a very appealing field as it addresses the needs of millions of people worldwide. Various strategies have been developed and these include tissue engineering as potential methods to repair and regenerate bone and cartilage defects. Although various materials are under investigations, hydrogels are one of the most attractive biomaterials in bone and cartilage tissue engineering. This is due to their moderate biocompatibility, hydrophily, and 3D network structure. This review is a summary of current progress towards of stimuli-responsive hydrogels in bone and cartilage regeneration. The challenges and disadvantages of stimuli-responsive hydrogels and future applications are briefly described.

Overall, this is a good review article that require minor revisions before consideration for publication.

Specific Points

1.     Figure 3- what is chick chemistry? Revise. Click chemistry.

2.     There are many statements lacking References- when authors say ‘ it was demonstrated that….’. This statement requires a Reference at the end. Revise.

3.     Some of the Figures are difficulty to read. For example, Figure 7. Make sure text in Figures is readable.

4.     Overall, this is a good review article

Reviewer 2 Report

This manuscript studied about the various hydrogel manufacturing techniques and applications.

However, this study seems to be more suitable as a text book for general undergraduates rather than a review for researchers or investigators. Compared to the review papers of hydrogel related to this manuscript, it is necessary to limit the subject and analyze it in depth.

Many review papers on hydrogel have been published by related researchers.

-        (Supramolecular Hydrogelators and Hydrogels: From Soft Matter to Molecular Biomaterials. Xuewen Du, Jie Zhou, Junfeng Shi, Bing Xu, Chem Rev. 2015 Dec 23; 115(24): 13165–13307.

-        In situ Forming Injectable Hydrogels for Drug Delivery and Wound Repair. Robert Dimatteo, Nicole J. Darling, Tatiana Segura, Adv Drug Deliv Rev. 2018 Mar 1; 127: 167–184

-        Irreversible and Self-Healing Electrically Conductive Hydrogels Made of Bio-Based Polymers. Ahmed Ali Nada, Anita Eckstein Andicsová, Jaroslav Mosnáček, Int J Mol Sci. 2022 Jan; 23(2): 842

The previous papers were clearly compared and analyzed in a table by comparing the papers of several researchers with each other.

In this manuscript reported the research status of hydrogels aimed at bone and cartilage regeneration. In this review, various hydrogel types were classified, and the research contents were mentioned through various references for each subject. Therefore, it is necessary to distinguish it from other previous review papers.

-        It is necessary to analysis the references for each section in a table and organize the comparison of previous studies at a glance. For example, references should be compared with each other for on synthesis methods, materials (natural, synthesis, hybides), gelation time, degradation time, sustain release, chemical structural formulas, cytotoxicity, or clinical application problems. And among the above items, references should be analyzed and organized in a table, focusing on one or two items, and compared with each other.

-        There is a difference in the amount of references for each subject (There are differences in the amount of reference papers, from 3 to 20, for each topic). The more reference to the pH-responsive hydrogel, ROS-responsive hydrogel, Photo-responsive hydrogel sections is needed. A uniform review is needed for each section.

-        The figure shows the results of a study on one reference. It is recommended to compare and analyze the commonalities and differences of various references and to schematize or diagram them.

Reviewer 3 Report

Dear Authors, Dear Editor,

In this manuscript Li and co-workers review the literature regarding “Applications of Stimuli-Responsive Hydrogels in Bone and Cartilage Regeneration”.

I must congratulate the authors for their efforts in navigating the extensive literature in this area and for arriving at a meaningful selection of the most relevant articles.

The review is well planned and written, and most importantly, the information is well discussed and integrated.

The length and breadth of the review and the number of references cited seems to me appropriate.

This review is likely to attract a great deal of attention from researchers in the nanomedicine and tissue engineering areas owing to the importance of bone and cartilage regeneration in the clinical context.

This work is clearly worth publication.

Some points must be made clear before publication:

- This review deals mostly with polymer-based cross-linked hydrogels. A “parallel/equivalent” review could probably be written on self-assembled hydrogels based on low molecular hydrogelators, e.g. peptides. The authors should include in the manuscript a clear statement on the scope of their review regarding the type of hydrogels.

- The authors should summarize the information on a table format for allowing for a quick comparison of the different systems.

- The quality of some Figures/Schemes must be improved regarding especially their size, e.g. Figure 2A.

- The authors should include an illustrative Figure in the “Electro-responsive Hydrogels” section.

- The structure of peptide P11-4 (CH3COQQRFEWEQQFEQQNH2) is some how ambiguous. Is there a acetyl group on the N-terminal? Please specify the sequence of peptide P11-4.

Best regards

Reviewer 4 Report

Comments:

1.     This is an interesting paper, however the authors should provide some future outlook for the stimuli-responsive hydrogels. Is that any existing limitations and what improvement can be done for future work.

Reviewer 5 Report

Some responsive hydrogels applicable in bone and cartilage tissue engineering have been reviewed. The subject is valuable and interesting. Hence the manuscript is potentially publishable. However, there are some points which should be addressed and discussed in the revised version for further completion of the review as mentioned below:

1.       Graphene sheets [Graphene nanogrids for selective and fast osteogenic differentiation of human mesenchymal stem cells], graphene hydrogels [Application of Graphene Oxide-Based Hydrogels in Bone Tissue Engineering] and graphene foams are known as one of the suitable materials for tissue engineering including ontogenesis and bone regeneration. This subject should be mentioned and discussed in the revised version as a (sub-)section.

2.       It has been mentioned that “Tissue engineering is a novel strategy to promote tissue repair and regeneration by combining related supporting cells, three-dimensional scaffolds, and bioactive factors”. This can be supported by, e.g., [Prevascularized Micro-/Nano-Sized Spheroid/Bead Aggregates for Vascular Tissue Engineering].

3.       The cellulose nanofiber scaffolds are considered as one of the suitable materials in drug loading and tissue engineering. See, for example, [Regenerated cellulose nanofiber reinforced chitosan hydrogel scaffolds for bone tissue engineering] and [An injectable and self-healing cellulose nanofiber-reinforced alginate hydrogel for bone repair]. This should be mentioned and discussed in the revised version.   

4.       It has been mentioned that “Previous studies have demonstrated that Electric-responsive hydrogels can mediate cell adhesion, proliferation, migration, and growth from different tissues, including cardiac, skeletal, muscular, and neural tissues”. This can be further completed as follows: “Previous studies have demonstrated that Electric-responsive hydrogels can mediate cell adhesion, proliferation, migration, and growth from different tissues, including cardiac [Electrically conductive carbon-based (bio)-nanomaterials for cardiac tissue engineering], skeletal [Hydrogel-Based Fiber Biofabrication Techniques for Skeletal Muscle Tissue Engineering], muscular [Muscle Tissue Engineering Using Gingival Mesenchymal Stem Cells Encapsulated in Alginate Hydrogels Containing Multiple Growth Factors], and neural [Rolled graphene oxide foams as three-dimensional scaffolds for growth of neural fibers using electrical stimulation of stem cells] tissues”.

5.       Could the authors discus about the role of MOFs in responsive hydrogel and bone tissue engineering? See, for example, [Tunable and Controlled Release of Cobalt Ions from Metal–Organic Framework Hydrogel Nanocomposites Enhances Bone Regeneration].

Round 2

Reviewer 2 Report

This manuscript systematically analyzed and review for various type and research trends of hydrogel. The author clearly summarized and analyzed the reference on the research trends.

It is believed that it can be a good guide for hydrogel researchers in the future.

Reviewer 3 Report

Dear authors, Dear Editor,

I must congratulate the authors for their efforts in addressing not only my suggestions but also the comments of the other referees. I believe that the manuscript has now reached the publication standard.

Best regards

Reviewer 5 Report

The manuscript has been revised well and can be considered for publication.